# Quasi-Movements and “Quasi-Quasi-Movements”: Does Residual Muscle Activation Matter?

**DOI:** 10.3390/life13020303

**Published:** 2023-01-21

**Authors:** Anatoly N. Vasilyev, Artem S. Yashin, Sergei L. Shishkin

**Affiliations:** 1MEG Center, Moscow State University of Psychology and Education, 123290 Moscow, Russia; 2Department of Human and Animal Physiology, Faculty of Biology, M.V. Lomonosov Moscow State University, 119234 Moscow, Russia; 3Faculty of Philosophy, M.V. Lomonosov Moscow State University, 119991 Moscow, Russia

**Keywords:** quasi-movements, motor imagery, movements, attempted movements, sensorimotor rhythms, intention, EEG, EMG, brain-computer interfaces, BCI

## Abstract

Quasi-movements (QM) are observed when an individual minimizes a movement to an extent that no related muscle activation is detected. Likewise to imaginary movements (IM) and overt movements, QMs are accompanied by the event-related desynchronization (ERD) of EEG sensorimotor rhythms. Stronger ERD was observed under QMs compared to IMs in some studies. However, the difference could be caused by the remaining muscle activation in QMs that could escape detection. Here, we re-examined the relation between the electromyography (EMG) signal and ERD in QM using sensitive data analysis procedures. More trials with signs of muscle activation were observed in QMs compared with a visual task and IMs. However, the rate of such trials was not correlated with subjective estimates of actual movement. Contralateral ERD did not depend on the EMG but still was stronger in QMs compared with IMs. These results suggest that brain mechanisms are common for QMs in the strict sense and “quasi-quasi-movements” (attempts to perform the same task accompanied by detectable EMG elevation) but differ between them and IMs. QMs could be helpful in research aimed at better understanding motor action and at modeling the use of attempted movements in the brain-computer interfaces with healthy participants.

## 1. Introduction

Quasi-movements (QM) are “volitional movements which are minimized by the subject to such an extent that finally they become undetectable by objective measures” [1]. The following procedure was introduced by Nikulin et al. [1], who first described this phenomenon, to teach experiment participants performing QMs: Participants are told to perform thumb abductions with gradually decreasing amplitudes. They are not directly observing the movement but are encouraged to minimize it until the related electromyography (EMG) signal monitored by an experimenter becomes indistinguishable from the background level. The participants are not aware of the lack of actual movement and its EMG signs and continue their cognitive-motor action, which now is the QM. 

Nikulin et al. [1] reported that QMs were accompanied by the suppression (event-related desynchronization, ERD) of the electroencephalography (EEG) sensorimotor rhythms, a well-known correlate of cortical activation related to motor actions. The topography of the ERD during a QM task was similar to what was observed during overt movement execution and kinesthetic motor imagery, but its amplitude was lower than in overt movement and higher than in imagery. Thus, QM was considered as a part of the continuum, including overt movements and motor imagery [1].

Since no correlation was found between the EMG root mean square (RMS) and contralateral EEG effects under both motor imagery and QM [2], the stronger sensorimotor rhythm ERD under QM could not be attributed to higher muscle activation in QMs, but instead should be considered as a reflection of more qualitative differences between the two tasks. Differences between QMs and imaginary movements (IM) observed for the repetition suppression effect [2] also suggested that their mechanisms may differ not only quantitatively but also, to some extent, qualitatively. 

Notably, QMs were easy to learn, and the simulated brain-computer interface (BCI) performance was higher [1] and more stable during repeated movements [2] for QMs compared with the kinesthetic motor imagery, suggesting prospects for their use in BCI applications. 

QMs are also similar to attempts to make a movement by a person who cannot make it due to paralysis or amputation. Such “attempted movements” can be used to control a BCI, similarly to the control via motor imagery, sometimes even more effectively [3,4]. However, it is difficult to study the attempted movements in healthy participants, because (temporal) artificial paralysis is needed for such studies. Thus, one more possible application of QMs could be their use as a model for attempted movements in the experiments with healthy participants [1]. 

Finally, QMs could enable experimental studies of the brain machinery for movement control without contamination from the peripheral feedback [1].

The phenomenon of QMs has been explored so far in a very limited number of studies [1,2,5,6,7,8]. One of the remaining open questions related to it is whether the higher contralateral ERD of the EEG sensorimotor rhythms in QMs relative to IMs is related to higher muscle activation. Zich et al. [8] in a study, which mostly followed the methodology by Nikulin et al. [1], did not confirm the higher ERD and better BCI accuracy in the QM task compared with the kinesthetic motor imagery task. Moreover, while Nikulin et al. [1] reported no significant difference in the muscle activation between the motor imagery and quasi-movements, they did not analyze the distributions of trial EMG values in detail, mostly relying on the time-average along a trial. They also widely used visual inspection, according to which few QM trials had occasional signs of task-related muscle activation. If the procedures they used for the EMG analysis underestimated the rate of QM trials with elevated task-related EMG, which we will also call “quasi-quasi-movements” (QQM), this could, in turn, cause the underestimation of the correlation between the EMG and the EEG effects. Therefore, it could not be ruled out that the difference between the EEG effects from the kinesthetic motor imagery and the QM is based only on the difference in the amount of muscle activation. 

A more precise analysis of the possible relation between muscle activation and sensorimotor cortex activation observed under the QM would be also helpful for better understanding of their relation to kinesthetic motor imagery and attempted movements, and for better understanding of the motor action in general.

Here, we re-examined the relation between the residual muscle activation during the execution of QMs and the brain activation. We asked our participants to perform movements in response to sounds that were presented in triplets with relatively short interstimulus intervals. We expected that this instruction would have the following effects: (1)The QM execution would be somewhat more difficult than with the original protocol (where the abductions were self-paced), so that a significant amount of EMG would be produced in a higher number of trials.(2)The ERD onset would have a lower time jitter between trials, which would help to compare the early brain activation related to QM onset between trials with and without the EMG production.(3)Using triplets instead of six movements in a row (as in Nikulin et al. [1] study) could minimize the repetition suppression effect [2], thus enabling more precise ERD measurements.

For EMG quantification, we used procedures enabling the detection of relatively short and low-amplitude muscle activations (but, following Nikulin et al. [1], still filtering off very transient muscle activations, which may last just for milliseconds and cannot lead to overt movements), indicating that a movement partly deviated from what is understood as QM. It also might be possible that participants could consciously perceive such movement attempts as different from QMs without muscle activation, and that stronger ERD would reflect their reaction, e.g., to the possible error in following instructions. We also made our ERD/ERS analysis more sensitive to the short-term signs of brain activation. 

We expected that if any muscle activation is related to the qualitative change in the studied phenomenon, namely, turning QMs into overt attempts to make a movement, the sensorimotor ERD amplitude would correlate with the EMG amplitude, and the ERD time courses for higher and lower EMG trials will differ even before the participant could receive proprioceptive feedback or detect an error in following instructions. 

In this study, we indeed found more EMG and stronger ERD/ERS in the QM condition compared to the IM. However, these effects of conditions were independent of each other, implying that the stronger ERD/ERS could not be caused, e.g., by efferent effects from the muscles. Thus, the difference between QMs and IMs can be more profound than it was considered.

## 2. Materials and Methods

### 2.1. Participants

In total, 23 naïve right-handed, healthy volunteers (12 males and 11 females, aged 22.5 ± 2.7 years, M ± SD) participated in this study. Two of them were left-handed, according to their self-reports. All of them reported no neurological diseases and had normal or corrected to normal vision. One more participant only took part in the first session of the experiment without the EEG recording and was excluded because the maximum allowed interval between the sessions (21 days) had been exceeded. All participants were introduced to the procedure and signed informed consent. The experimental procedures were approved by the local ethics committee and were in agreement with the institutional and national guidelines for experiments with human subjects, as well as with the Declaration of Helsinki.

### 2.2. Experiment Design

The experiment consisted of two sessions (Figure 1), with a gap between them lasting from 7 to 17 days (8.5 ± 2.8, M ± SD). During Session 1, participants learned three skills, in a fixed order: full-amplitude thumb abductions (overt movement, OM), the related quasi-movement (QM), and the kinesthetic motor imagery of thumb abduction (imaginary movement, IM). After learning a new skill, participants consolidated it through multiple repetitions of the movement. Finally, the participants practiced a mix of QM and IM trials, to better understand the difference between them. 

The EEG was recorded only in Session 2 (the main session), where the participants reproduced the learned skills. In this session, OM again was the first condition, but then QM and IM were presented in random order, which was counterbalanced over the group. 

Between the conditions, participants were surveyed (see Section 2.6), which took about two minutes. After the second condition, participants were also asked about possible fatigue, and a five-minute break was provided if fatigue was reported. Breaks between blocks within conditions typically took between 15 and 45 s. The full experiment duration (including filling the informed consent, electrode mounting, surveys, etc.) was typically 70–80 min for Session 1 and 140–180 min for Session 2. 

The movement had to follow an auditory rhythm (triplets of sounds), which were either periodic or aperiodic (see details below, in Section 2.3.1). In the main part of the experiment, trial blocks with periodic and aperiodic rhythms were presented within each condition in an order which was fixed for a participant, while randomized and counterbalanced over the group. After the main part of the QM condition, an additional, shorter sub-condition of self-paced QM was run. The EEG was also recorded in eyes open (one minute) and closed (one minute) resting states in the beginning and the end of Session 2. The data from self-paced sub-condition and resting-state recordings were not analyzed in this study.

### 2.3. Procedure

Participants were seated in an armchair in front of a table with a computer screen on it. The right forearm was comfortably placed on a thick, soft material over a horizontal hard surface positioned near the armchair with its palm turned left (inward). 

Thumb abduction was used following the previous QM studies [1,2,5,6,7,8]. This movement is made by contractions of the m. abductor pollicis brevis, a muscle well-suited for QM studies, as it is superficially located and can be well monitored with the surface EMG [1].

Unlike in the previous QM studies, where the movements were executed with both hands, we asked the participants to make all movements using their right hand only, because, unlike these studies, our study did not examine two-class BCI classification accuracy, but instead was focused on the EMG and EEG power changes and their time courses. 

Another difference was the use of triplets of sounds for pacing the movements. We expected that when participants were asked to make movements following sounds, the EMG and ERD responses would be more stable in time, allowing for more sensitive quantification of both EMG and EEG effects. From a practical perspective, it is interesting to explore how well the EEG changes associated with QM and IM, which follow some fixed temporal patterns, can be detected by a BCI classifier (this is being performed in another study using the data from this experiment). Two types of triplets were used, periodic and aperiodic (described in the next subsection).

Finally, to ensure a more stable EEG along the conditions and to provide baseline data for constructing spatial filters and (in a separate study) for BCI modeling, we used, following Vasilyev et al. [9,10,11], a visual attention task (visual task), which was presented among movement trials. Due to the engaging nature of this task, participants were expected to effectively switch from their focus on motor activities in the main conditions without falling into mind wandering. Note that in the EEG analysis here, we focused on the sensorimotor components extracted from the EEG using spatial filters, so we did not expect that the task-related variations in visual EEG components could significantly affect the results, due to the very distinct localization of those components.

#### 2.3.1. Block and Trial Structure

As shown in Figure 1, five pairs of blocks were presented per condition (OM, QM, or IM). Within a pair, one block was with periodic rhythm and one with aperiodic rhythm. A block consisted of five trial sequences. A trial sequence consisted of two movement trials and one visual task trial, with varying length intervals for rest between them (Figure 2).

In a movement trial, an image of the right hand appeared on the computer screen for 3 s. Starting 500 ms after the image onset, a rhythmic sequence of three identical percussion sounds (duration 13 ms each) was played at a comfortable intensity level via a loudspeaker column (“soundbar”) (Stage Air 1.0, Creative, China), located under the screen. Stimulus onset asynchronies (SOAs) were equal (600 ms) in the periodic rhythm and different (400 ms and 800 ms) in the aperiodic rhythm. The participant made an OM, QM, or IM after each sound, trying to keep up with the rhythm. The interval between trials randomly varied from 2 to 4 s with uniform distribution (in the same way within and between sequences).

In the visual task, a picture with complex geometrical shapes, same as that used for similar purposes in motor imagery studies by Vasilyev et al. [11], was presented on the computer screen for 4 s. Like in these studies, a participant had to silently count any elements of the picture of the same type (e.g., lines, dots, angles, protrusions) at their own comfortable pace. 

### 2.4. Session 1

#### 2.4.1. Practicing Overt Movements (OM)

The first session was designed for learning the OM, QM, and IM. A participant first learned the overt right thumb abduction movement from the experimenter (who demonstrated the movement on himself) and tried to reproduce it. Then, the participant practiced the movement over 30 trials following the rhythms (each trial included 3 sounds, as in other parts of the experiment, so 90 movements were executed; however, no visual task was used at this stage). If the participant needed more practice, more blocks were added. The experimenter supervised the participant so that they executed the correct movement.

#### 2.4.2. Practicing Quasi-Movements (QM)

The EMG electrodes were applied, and the right hand was covered with an opaque case (not touching the hand) for this and further conditions.

The procedure for learning QMs mostly followed Nikulin et al. [1]. In order to avoid doubts about whether movements were actually executed, QMs were called “micromovements” in all communications with the participants. Initially, the EMG was displayed on the screen in front of the participant. The participant verified that the signal on the screen corresponded to their muscle activity by moving their thumb. The experimenter then told the participant to continue self-paced abduction movements while decreasing their amplitude and the EMG signal amplitude. He notified the participant that the sensation of the movement might vanish at low amplitudes.

When the movement-related EMG increases became barely noticeable, the experimenter moved the window with the EMG signal to his screen (out of view for the participant) and told the participant to continue executing “micromovements,” but this time following the rhythms. At this stage, the visual task was also added after each two QM trials, so now the trial sequence was the same as in further parts of the experiment. The experimenter warned the participant that he could ask them to decrease the amplitude of the movement if the EMG response was greater than it should be. Now, when the experimenter saw a discernible EMG response, he asked them to make the movement even smaller until a stable absence of response was achieved. QMs were practiced over 4–6 blocks of trial sequences. As a block included 5 trial sequences, a sequence included 2 QM trials (followed by a visual task), and each trial included 3 sounds, which should be followed by the QM, the total number of QM attempts was (4–6) × 5 × 2 × 3 = 120–180. The practice typically ended when the participant could make rhythmic QMs regularly without a visible increase in their EMG signal amplitude.

#### 2.4.3. Practicing Kinesthetic Motor Imagery (IM)

The experimenter first asked the participant to close their eyes and execute a full-amplitude right thumb abduction several times at different rates, while attending to their sensations. Then, he asked them to verbally describe those sensations. Among all the descriptors, the experimenter chose the ones that did not specify visual or tactile content, for example, tension, weight, or heat. If the participant did not name these descriptors on their own, the experimenter obtained them via asking questions about the participant’s experience and the physical events occurring in the hand. 

Next, the participant was requested to imagine the selected sensations dynamically, as if they were performing a real thumb abduction. When the participant was confident enough in the quality of the motor imagery, they were asked to imagine the movement now following the rhythms with open eyes. Again, a visual task was added after each two trials with sound triplets. In total, 2–3 blocks of trial sequences were offered (10–15 trial sequences, i.e., 60–90 IM attempts). The number of IM trials was lower than in the QM practice, because the participants apparently learned to make IMs in a stable way much faster than QMs. Like in the QM practice, the experimenter observed the EMG signal and informed the participant if the IM regularly resulted in a discernable EMG amplitude.

#### 2.4.4. Improving Understanding of the Difference between QM and IM

Finally, a mixed practice was offered with the purpose to help the participant better feel the difference between the QM and IM. Here, along 4 blocks of trial sequences (with 120 sounds prompting movement attempts), at random moments, the experimenter gave a command to switch from QM to IM, and vice versa.

### 2.5. Session 2

In Session 2, the participant reproduced the skills learned in Session 1. Before the session, the experimenter mounted the EEG and EMG electrodes, and also covered the participant’s right hand with an opaque case. The participant’s EEG and EMG were registered in all conditions. 

The session consisted of three conditions: the OM, QM, and IM. Each of them started with 1 recap block. During the recap block, the participant could communicate with the experimenter. Before the QM condition, the number of recap blocks could be higher (up to 5 blocks) if the participant was not able to reproduce the QM phenomenon in the first block. The number of blocks could be increased on demand of the participant. Before the IM condition, the experimenter asked the participant to remember which feelings they should simulate during the IM. If the participant could not recall the descriptors, the experimenter would help them. 

In every condition, after the recap blocks, the participant executed the required activity in 10 blocks, with the rhythm switching after each block. It was in these blocks where we recorded the EEG and EMG. As in the first session, in each block, after every two action trials, a trial with the visual task occurred. The participant was reminded about the task before the session began. Apart from recap blocks and recording blocks, the QM condition included 2 blocks with self-paced QMs, but the analysis of these data is beyond the scope of this paper.

### 2.6. Surveys

After each condition, participants answered the experimenter’s questions orally in free form. They were also free to give additional feedback on every condition if they wanted to. All questions were asked in Russian, which was the native language of all participants; below we present their translations into English.

To ensure the correct performance, the following questions were asked after every condition, except the mixed one: “From your point of view: Were you able to execute the correct movement/ execute micromovements/imagine the correct movement?” (in Session 1 only); “Did you manage to execute the movements/execute micromovements/imagine movements following the rhythm?” (in both sessions); “Were you able to maintain concentration throughout the entire condition?” (in Session 2 only). The responses did not reveal significant problems, except for some loss of concentration in a few cases (if this was reported, additional rest was provided).

At the end of Session 2, participants were asked to answer questions to make sure that they correctly performed the visual task between the rhythmic trials: “Were you able to fully relax your hand while counting the elements in the picture?”; “How many elements did you count on average per picture presentation?” Again, significant problems were not revealed.

To find out what is specific to the subjective experience in QMs, the experimenter asked additional questions after this condition: “Did you feel that you were moving your thumb when you executed the micromovements?”; “Did you feel you were tensing a muscle when you executed the micromovements?” After the QM and IM conditions, the following question was also asked: “To what extent were the actions you performed in this task imaginative or real?” At the end of Session 2, participants were also asked: “Which was more difficult: imagining the movement or performing micromovements”.

At the end of both sessions, the participant was also requested to give two reports: “Describe the subjective difference between regular movements and micromovements.”; and “Describe the subjective difference between imagining the movement and micromovements.” These reports will be analyzed in a separate publication specifically focused on the subjective experience in the QM.

### 2.7. Signal Acquisition

EEG and EMG signals were recorded at 1000 Hz sampling rate and 24-bit voltage resolution with 0…300 Hz passband using the *NVX136* DC EEG amplifier (*Medical Computer Systems*, Moscow, Russia). The EEG was acquired at 128 locations, with Ag/AgCl ring electrodes positioned according to the 10–5 system at the *MCScap Professional* 128 electrode cap (*Medical Computer Systems*, Moscow, Russia). The EEG was obtained and stored as single-ended signals with the ground electrode at the Fpz position. As we did not analyze the EEG channel wise beyond the bad channel detection and interpolation (the main EEG analysis was run solely on components), we did not use a reference channel. EEG electrode impedance was kept below 20 kΩ. EMG electrodes were placed on the right palm, one electrode over the belly of the *m. abductor pollicis brevis* and the other over the proximal base of the phalanx. The signal was re-computed to a single bipolar EMG channel, both for online visualization and for the processing. The EMG electrode impedance was kept below 50 kΩ. 

Signal recording and stimuli presentation were organized using Resonance [12], a software platform for multimodal real-time experiments. The time of the visual stimuli presentation was monitored with a photo sensor located in the left low corner of the screen, whose signal was recorded along with the EEG and EMG. As acoustic and visual stimuli were presented using a video stream, allowing for precise measuring asynchrony between them, the photo sensor signal also allowed for the precise estimation of the acoustic stimuli timing in EEG/EMG time coordinates.

### 2.8. EMG Processing

EMG data were filtered offline in the 5–150 Hz frequency band with the 4th order zero-phase Butterworth filter, and a 50 Hz notch filter. EMG amplitudes were normalized (divided) by the SD of EMG amplitudes from the IM condition of the same participant (from both movement trials and visual task trials, including time intervals between trials but excluding intervals between the blocks, as well as the first 3 s and the last 3 s from each block; IM and visual task data were used because they were supposed to have the lowest rate of muscle activation). We did not remove at this step any trial data (note that in the IM trials, the substantial EMG amplitude increases were not observed). RMS values were computed in 25 ms windows with 24 ms overlap. The baseline was computed separately for each trial as an average of the RMS values from two intervals of 750 ms duration, one before and one after the movements, specifically, at −300…+450 ms and +2100…+2850 ms (here and further, time relative to the onset of the visual cue is given; see Figure 3 for the temporal positions of the time intervals within the movement trials used in the analysis). It was subtracted from all RMS values, providing the corrected EMG RMS values. The time interval for characterizing movement-related EMG was defined as +500…+2000 ms from the visual inspection of the trial-averaged EMG in the OM condition (the interval where pronounced EMG was found, see Figure 4A). In most analyses, including dividing the trial set into “low EMG” and “elevated EMG” trials, the 95th percentile of the corrected EMG RMS values from this interval was used, denoted as $\bar{rms}$, or (in the statistical models) as *peakRMS*. 

### 2.9. EEG Processing

#### 2.9.1. Setting Individual Frequency Bands and Spatial Filters

To detect bad EEG channels, the channel power spectral density was computed using Welch’s method (using the MATLAB *pspectrum* function with default parameters) for all data from a participant, and channels which differed from the all-channel median by 5 median absolute deviations were visually inspected. Such channels (P3, CPP5h, CPP3h, P1, P5, PPO5h) were found only in one participant. They were replaced with interpolated signals using EEGLAB’s [13] *interp* function with the “spherical” option.

Sensorimotor rhythms were extracted from the EEG using spatial and frequency filtering. Given the high interindividual variability of the sensorimotor rhythms, both types of filtering were individually tuned for better separation of the relevant components from the background EEG. Spatial filtering was performed using generalized eigen-decomposition (GED) on covariance matrices (also known as Common Spatial Pattern, CSP, in the BCI literature), which was shown to effectively extract functionally relevant components with different spatial patterns from the EEG [14,15,16]. Due to the robustness of the covariance estimation (see below), no prior rejection of epochs with artifacts was used. Frequency filtering was used to extract the most reactive part of the frequency spectrum within the alpha and beta bands. 

Tuning both the spatial and frequency filtering was based on the contrast between the OM and the visual task, as we expected that this contrast would most clearly represent the spatial and frequency pattern of the relevant sensorimotor EEG components. Note that Nikulin et al. [1] found no significant difference in the ERD/ERS topography between the OM and QM and between the OM and IM for the same thumb abduction task as we used here. The visual task data for this analysis were taken from the trial sequences of the same OM condition. Thus, the frequency and spatial filter tuning was fully based on the EEG, which was not used in the statistical analysis (which was applied to the data from the QM and IM conditions).

The following pipeline was applied separately to each participant’s 128-channel EEG data to define the individual spatial and frequency filters:Bandpass filtering in 6–30 Hz range using a zero-phase FIR filter with 40 dB suppression and 1 Hz-wide transition band.Extraction of 1.5 s OM epochs related to movement (from the first sound onset to 500 ms after the last sound onset, i.e., +500…+2000 ms relative to the beginning of the trial, which was at the onset of the visual cue, see Figure 3) and 3.5 s epochs related to the visual task (+500…+4000 ms relative to the visual cue onset).Covariance matrices were estimated separately for concatenated epochs related to the OM ($C_{move}$) and to the visual task ($C_{visual}$) using the MATLAB *robustcov* function employing the robust Olive-Hawkins method [17].Generalized eigenvectors and corresponding eigenvalues were calculated, respectively, as columns in W and as Λ in Equations (1) and (2):(1)$$\Lambda =argmax\left\{\frac{W^{T}C_{move}W}{W^{T}C_{visual}W}\right\}$$
(2)$$C_{move}^{-1}C_{visual}W=W\Lambda$$

Eigenvectors were sorted according to Λ values. Then, the first and the last eigenvectors in *W* had the lowest and the highest values of $C_{move}^{}$ to $C_{visual}$ ratio, respectively.

5.Defining the individual alpha band started from extracting the spatial EEG components demonstrating ERD in the OM condition. Individual spatial filtering was performed by multiplying the EEG from all conditions by the first three components, W(:, 1 :3). (The reason to take three components was that at least two components were needed to represent contra- and ipsilateral activation, and sometimes an irrelevant component could occur amongst the first two). 6.Wavelet transform (Morlet wavelet with varying number of cycles—the “superlets”, using the procedure proposed by Moca et al. [18]) was applied to the EEG components extracted at step 5. The resulting spectrograms were averaged separately over the OM and visual task trials (for the latter, interval +1000…+3000 ms was used), giving *P_OM_* and *P_visual_*. The normalized spectrogram was obtained as 10 × log (*P_OM_*/*P_visual_*). In total, the 6-bin moving average was applied to produce the 0.5 Hz frequency resolution. The individual alpha band was defined as ±1.5 Hz from the peak value of the component with contralateral activation, as its spectrum was best defined and the difference from the component with ipsilateral activation did not exceed 0.5 Hz.7.Defining the individual beta band started from extracting spatial EEG components corresponding to the event-related synchronization (ERS, i.e., power increase relative to the baseline in a given frequency band) in the OM condition, such as at step 5, but using the last three spatial components, W(:, 126 :128). 8.Wavelet transform (same as in step 6) was now applied to the EEG components extracted at step 7. The beta band (both left and right borders, allowing the bandwidth to vary) was defined manually based on visual inspection of the periodograms, so that it covered the highest synchronization. The manual procedure was needed in this case because high individual variability was observed for the ERS onset (it was observed at approx. +2.5…+3.5 s relative to the visual cue, i.e., approx. 2.0–3.0 s after the start of the movements) and especially for the bandwidth (which could be anywhere from 6 to 13 Hz in this study).9.Steps 1–4 were applied to the raw EEG data using (at step 1) the individual frequency bands estimated at steps 5–8, resulting in two sets of spatial filter sets, $W_{alpha}$ and $W_{beta}$, which were expected to include filters more precisely tuned to, respectively, the alpha and beta components of the sensorimotor rhythms.10.These precisely tuned spatial filters were then identified using the following procedure, designed on the basis of recommendations by Cohen [14] and Muralidharan et al. [19] and our experience from the studies [11,20]. First, weights approximating the forward models of the sources [21] were computed as *A* ≈ *W^−T^* ≈ *W^T^*$C_{move}$. Secondly, scalp topographies of the weights in A were visualized for each component, and individual components in the alpha and beta bands were selected by their visual inspection on the criteria that smooth but spatially limited maxima were observed in the sensorimotor areas. Specifically, two alpha band components with maxima in the location ranges FC5..CP1, FCP6..CP2 (the contralateral and ipsilateral sensorimotor areas) were selected from the first five components representing the alpha band ERD ($W_{alpha}$), and one beta band component with maxima in the location ranges F5..CPz (here, only the contralateral area, because no components were found in the ipsilateral cortex) was selected from the last five components representing the beta ERS ($W_{beta}$). Almost all selected alpha band components were consistent with the radial dipole orientation, while both radial and tangential configurations were seen among beta band components (Appendix A).11.Spatial filters for two alpha band components (corresponding to contralateral and ipsilateral alpha band ERD) and one beta band component (corresponding to contralateral beta band ERS) of the sensorimotor rhythm, identified in the previous step, were applied to the raw EEG.

#### 2.9.2. Time-Frequency Analysis and ERD/ERS Quantification

Time-frequency analysis was applied to the three EEG components of the sensorimotor rhythm obtained according to the above-described algorithm for each participant. This analysis was based on the wavelet transform and used the Morlet wavelet with varying number of cycles, as proposed by Moca et al. [18]. Wavelet scales varied from 3 to 35 Hz with 0.5 Hz step. Power values were converted into dB with normalizing by the average over 0.5 s time interval preceding the visual cue (−500…0 ms) and averaged over the individual frequency bands (defined at steps 6 and 8 of the above algorithm). Using these average values, the peak power was computed for each trial as the 10th percentile of the alpha band power in +500…+2000 ms time interval for ERD (Figure 3) and as the 90th percentile of the beta band power in +2000…+3500 ms interval for ERS (not shown in Figure 3), in the EEG components representing ERD and ERS, respectively. The reason to use peak ERD/ERS power values was that they represent the highest activation/suppression, which can be achieved at different times within the trial, due to trial-to-trial variations of various factors. 

In addition, the pre-trial mean power was computed for each component over the −1000…−250 ms time interval relative to the visual cue onset, to be used in the statistical models (next subsection) as a kind of baseline. Note that the baseline intervals differed from what was used for the EMG quantification (Figure 3): the EEG baseline should be set early enough to avoid the influence from preparatory processes, while for the EMG the calmest intervals were expected to be relatively close to the movement.

### 2.10. Statistical Analysis

Linear mixed-effects models (LMM) were used to evaluate the effects of various factors on single-trial peak power values (dependent variable), separately for each of the three components of the EEG sensorimotor rhythm in each of the covert movement conditions (QM, IM). LMMs are a powerful statistical approach enabling accounting for within-subject covariates and for both individual and interindividual variance, even an under imbalanced design and with missed data. They well-fit the need of advanced EEG data analysis [22] and were recently successfully used in a number of neurophysiological studies (e.g., [23,24]). 

The following model was used to evaluate the effects of the movement type, EMG level, pre-movement power of the same EEG component, and the order of conditions (written in Wilkinson notation [25]):*peakPower* ~ 1 + *prePower* + *recN* + *trialN* + *prevT* + *cond*peakRMS* + *rhythm* + (*cond-1|subj*),(3)
where *peakPower* is the dependent variable (single-trial peak ERD or ERS power values for a given sensorimotor EEG component, see Section 2.9.2), the independent variables are listed in Table 1, “*cond*peakRMS*’’ corresponds to fixed effects *cond* and *peakRMS* and their interaction *cond:peakRMS*, while “*(cond-1|subj)*” stays for a random slope for conditions for each participant.

Since *peakPower* was normalized for each subject, the random intercept was not modeled. A few trials per model with anomalously high normalized residual variances (>3.5) were discarded prior to the final computation. The significance of the effects (i.e., of the related coefficients of the model) was evaluated using the *t*-test for individual coefficients or the *F*-test for groups of coefficients. The normality of residuals was controlled using the Shapiro-Wilk test (alpha = 0.01), as well as with visual inspection QQ-plots, and heteroscedasticity was also controlled by the inspection of “residuals—response” plot.

All computations were run using MATLAB 2022a, in some cases using functions from EEGLAB, ver.2021.1 [13].

## 3. Results

### 3.1. Survey Results

After the QM condition in both sessions, we asked participants if they felt any thumb movement during the QM. In the first session, only two participants reported affirmatively, and eight participants experienced this sensation only during part of the trials or to a very small extent. Thirteen participants reported that they did not feel any thumb movement. During the second session, there was an increase in the number of participants who reported feeling movement, with seven participants claiming it (Table 2, column 2). In total, 4 participants felt movement in part of the trails, and 12 did not feel any thumb movement.

In addition, 15 participants were asked during each session whether they felt muscle tension when they performed the QM. This question was introduced during the course of the experiments after one of the participants reported feeling tension in the absence of thumb movement. During the first session, only three people responded that they did not feel any muscle tension, one responded that they felt tension in part of the trails, and eight confirmed that they generally felt tension while doing QMs. During the second session (Table 2, column 3), three people felt no tension, three felt tension occasionally, and nine felt tension regularly. Interestingly, of these nine, only two also felt the thumb movement. 

After the QM condition in the first session, when asked if the QMs were real or imaginary, six participants responded that they were imaginary or more imaginary. Four participants reported that the QMs were equally real and imaginary. In total, 13 participants, more than half of the sample, stated that the QMs were real or more real than imaginary. During the second session (Table 2, column 4), 3 participants considered the QM to be imaginary or more imaginary, 2 were unsure, and 18 acknowledged them to be real or more real. In response to the same question about the IM, only one participant doubted whether the represented movements were imaginary: all the others either considered them completely imaginary or mostly imaginary. In general, the participants’ confidence in the reality of the QM increased during the second session, which is consistent with greater confidence in the presence of the feeling of movement.

When asked, at the end of the second session, which activity, QM or IM, was more difficult to perform, 11 participants responded that it was more difficult to perform the IM, 10 participants found the QM more difficult, and two responded that the QM and IM were equal in terms of difficulty.

### 3.2. EMG

Nikulin et al. [1] reported that both the pre- and post-stimulus (i.e., recorded before and during task performance) mean RMS EMG in their experiment were ~2 μV in both the kinesthetic motor imagery and quasi-movement tasks. They also noticed that this level was smaller or the same as that observed at rest for the same muscle, m. abductor pollicis brevis, by other authors [26,27]. In our data, the mean RMS EMG under the QM and IM were almost the same, 3.2 and 2.4 μV, respectively. The mean peak RMS EMG values in the Nikulin et al. [1] study were ~30 μV for both imagery and quasi-movement tasks, compared with ~340 μV for the overt movements. In our experiment, similar values were observed, but with higher contrast between the tasks: 405 μV for OM, 22 μV for QM, 7 μV for IM. 

Figure 4 presents the EMG RMS time courses. As expected, the EMG was indistinguishable from the background in the IM condition (Figure 4C), but not in the QM condition (Figure 4B). A sound-related (supposedly, movement-related) modulation was observed in the OM and QM conditions (Figure 4A,B). The first and the last EMG peaks were shifted in time relative to the sound onsets both in the OM and, even stronger, the QM condition. The shifts suggested that the participants reacted to the first bit (i.e., literally “following it”, as suggested by the instruction), but mostly tended to make the last movement before the last bit, i.e., based on the anticipation (similar to what is observed for the overt movement synchronization with complex rhythms, e.g., [28,29,30]).

However, the mean RMS could be insufficiently sensitive to short EMG bursts, while the peak RMS data could be too noisy. Therefore, further EMG analysis was mostly focused on another index, which we call here $\bar{rms}$ (the 95th percentile of the corrected EMG RMS values from the interval, where most EMG was observed in the OM condition, i.e., +450..+2000 ms). 

Cumulative distribution functions (CDFs) for $\bar{rms}$ values in each condition (Figure 5) indicated that the EMG production was practically indistinguishable between the IM condition and the visual task, as well as between periodic and aperiodic rhythm patterns in each condition. CDFs for the visual task and the IM condition resembled the normal distribution, although the IM distribution had a relatively long right tail. For $\bar{rms}$ below 0.5, CDFs for the IM and QM EMG seemed to follow similar (while qualitatively different) distributions, but for $\bar{rms}$ above 0.5, the CDFs strongly deviated. The visual inspection of the RMS EMG traces in a significant number of selected trials revealed that the EMG could not be observed in trials with $\bar{rms}$ < 0.5, and could often be noticed in the case of $\bar{rms}$ > 0.5. The grand averaged EMG RMS time courses for three ranges of $\bar{rms}$ in the QM condition confirmed the lack of EMG deviation from the background (visual task) level for trials with $\bar{rms}\;$< 0.5 (top left subplot in Figure 5), while a deviation from the background level and a clear temporal pattern were evident already in the averaged trial data with the slightly higher EMG level, 0.5 < $\bar{rms}$ < 1.0 (top middle subplot in Figure 5). Therefore, 0.5 was set as a threshold to distinguish between the QM trials supposedly without and with the muscle activation (low EMG and elevated EMG trials).

In total, 75.63% of the IM trials and only 34.32% of the QM trials had $\bar{rms}$ < 0.5. Participants highly varied in the percentage of trials with the elevated EMG ($\bar{rms}\;$ > 0.5), and only seven of them were able to refrain from producing such EMG levels in more than 50% of trials in the QM condition (of the summed number over trials with periodic and aperiodic sound patterns; see Table 2). Nevertheless, each category of trials, with elevated ($\bar{rms}\;$ > 0.5) and low ($\bar{rms}\;$ < 0.5) EMG, comprised more than 20% of all QM trials in 15 out of 23 participants. 

We compared the rate of trials with the elevated EMG ($\bar{rms}$ > 0.5) between participants who answered “Yes” and “No” to the question “Did you feel that you were moving your thumb when you executed the [QM]*?” The median rate of such trials was 48% for the 7 participants who answered “Yes”, and even slightly higher, 56%, for the 12 participants who answered “No” (the difference was not significant, according to the Wilcoxon rank sum test: *T* = 67, *p* = 0.8). We did not obtain enough data to apply such analysis for the answers to the other two questions shown in Table 2; nevertheless, in the available data, higher rates of trials with elevated EMG did not tend to be strongly associated with answering “Yes” to these questions as well.

Individual RMS time courses are shown in Appendix A.

### 3.3. EEG

Individual frequency bands set at steps 6 and 8 of the tuning/processing pipelines (see “Setting individual frequency bands and spatial filters” in Methods) are presented in Appendix A. Mean group values were 10.41–13.41 Hz for the alpha band components (exhibiting ERD, i.e., movement-related suppression) and 16.55–24.95 Hz for the beta band components (which exhibited ERS, i.e., post-movement enhancement).

Individual scalp topographies for the activation matrices *A* of the components of EEG sensorimotor rhythm obtained at step 10 of the pipeline (corresponding to the individual spatial filters) are shown in Appendix A. Components for the contralateral alpha band ERD (*contra ERD*) were found in 22 of 23 participants, while the ipsilateral alpha band ERD (*ipsi ERD*) and contralateral beta band ERS (*contra ERS*) components were identified in smaller numbers of the participants, but still in their majority: in 16 and 20, respectively. No components with an ipsilateral beta band ERS were found in any participant. 

We excluded from further analysis (including the linear mixed-effect models) participants 5, 17, and 22, who had too low numbers of low EMG ($\bar{rms}\;$ < 0.5) QM trials (2%, 1% and 5%, respectively) and participants for whom the EEG components were not found (dependent on the type of components). The averaged activation scalp topographies for the three EEG components are shown in Figure 6. Note that frequency bands and spatial filters were set based on the analysis of the OM data, but were mostly used in the analysis of the covert movement (QM and IM) data. The individual and grand averaged topographies of the ERS (beta band) component suggest source dipole orientation, which is more parallel to the scalp surface compared to the ERD (alpha band) components.

The QM trials with especially high EMG, $\bar{rms}\;$ > 5.0, and IM trials with $\bar{rms}$ > 0.5 were rejected. After this procedure, the rate of IM trials with $\bar{rms}$ < 0.5 became 76.47%, almost unchanged compared to the original rate, but slightly increased for QM trials, up to 40.40%. In total, data from 19 participants (3597 QM and IM trials) were used for the contralateral alpha band components with ERD, data from 14 participants (2652 QM and IM trials) were used for the ipsilateral alpha band components with ERD, and data from 17 participants (3207 QM and IM trials) were used for the contralateral beta band components with ERS. 

Spectrograms for each movement-sensitive EEG component in the covert movement conditions, QM and IM, are shown in Appendix A. 

Group-averaged time courses of the EEG components for all movement conditions, together with the EMG RMS time courses, are presented in Figure 7. For the QM condition, they are shown separately for the trials with the different EMG levels, low ($\bar{rms}\;$ < 0.5, solid green line) and elevated ($\bar{rms}\;$ > 0.5, dashed green line). Individual time courses strongly varied between participants: see Figure 8 for the contralateral alpha ERD component, and Appendix A for another two components. 

### 3.4. Linear Mixed-Effects Models

LMMs were separately applied to model effects on the three dependent variables, corresponding to the three types of the EEG components, whose topographies and time courses were presented above. Note that the OM condition was not included in this analysis. Detailed results for each independent variable are presented in Appendix A.

The pre-movement power (variable *prePower*) in the respective frequency band (alpha or beta) also had significant effects on the ERD/ERS in all EEG components, with higher pre-movement power associated with stronger ERD/ERS. For all EEG components, the type of rhythm (*rhythm,* periodic or aperiodic) did not show significant effects. The recording and trial numbers (*recN, trialN*) were near significant (*p* ~ 0.05) for the two ERD components, and significant for the ERS component, indicating a weaker EEG response to the QM and IM with higher recording and trial number. The effect of the previous trial (*prevT*) was near significant (*p* = 0.043), with lower ERD in the second trial in a pair, only for the ipsilateral ERD component, while being not significant for the two other EEG components.

Condition effects (variable *cond*) were significant for all EEG components; more specifically, ERD/ERS was stronger in the QM condition than in the IM condition. In addition, for all EEG components, the *cond* effect significantly depended on participants, as CIs for their SD estimates of the random effect *(cond|subjName)* did not include 0. Specifically, the SD estimates and CIs were 0.164 and [0.104 0.260], 0.065 and [0.018 0.231], and 0.215 and [0.143 0.323] for the contralateral alpha band ERD, ipsilateral alpha band ERD, and contralateral beta band ERS, respectively.

Effects of the EMG on the ERD and ERS were estimated using *F*-tests for the equality to 0 of all model coefficients that included the *peakRMS* factor. In the case of the alpha band ERD, this effect was not significant for the contralateral components (*F*(3587, 1) = 0.8332, *p* = 0.3615), and significant for the ipsilateral components (*F*(2642, 1) = 12.42922, *p* = 0.0004), where ERD was higher with the stronger EMG. For the contralateral beta band ERS, no effect was found in the QM condition (*F*(3197, 1) = 2.0878, *p* = 0.1486), while stronger EMG was associated with stronger ERS in IM ‘‘peakRM’’ coefficient = 0.16303, *p* = 0.0004). However, the beta band ERS results should be considered with caution, because ERS was very low in both the QM and IM conditions (mean group value below 1 dB), unlike in the OM condition (see spectrograms, Appendix A). 

## 4. Discussion

Previous studies of quasi-movements (QM) [1,8] and theoretical reflections on them in the literature (e.g., [31]) suggested that this phenomenon should be considered as a part of a continuum between overt and imaginary movements (IM). One of its manifestations could be the intermediate strength of sensorimotor rhythm modulation in the QM, positioned between the IM and OM, which was confirmed in the current study. Such intermediate positions of QM ERD could be related to intermediate levels of the activation of the efferent system, or other underlying mechanisms that manifest themselves in the variations of the residual muscle activation. The current study provides strong arguments against such a view.

Here, we compared the precisely quantified EMG and the ERD/ERS of sensorimotor EEG rhythms between the QM and IM (more exactly, kinesthetic motor imagery). The average EMG level in these conditions was similar to what was observed in the previous QM studies [1,8]. Unlike in these studies, which relied on simpler approaches to EMG quantification, we found that the distribution of peak trial EMG RMS ($\bar{rms}$ values, computed as the 95th percentile of the baseline-corrected EMG RMS from 25 ms windows), highly differed between the QM and IM, as well as between the QM and a nonmotor, visual task (Figure 5). Specifically, many of the QM trials, 75.68%, but much less of the IM trials, 34.37% of all IM trials, had $\bar{rms}$ > 0.5. However, participants who felt that their thumb moved in the QM did not have a higher percentage of $\bar{rms}$ > 0.5 trials compared with participants who felt that their thumb did not move in the QM. Moreover, the sensorimotor EEG component most sensitive to movements, i.e., the contralateral alpha band ERD component, was not related to rms, according to the results of the LMM analysis and Figure 7. Two other sensorimotor EEG components demonstrated only a weak relation to rms, and their LMM coefficients were much lower than the LMM coefficients accounting for the type of movement (QM vs. IM). The type of movement, independently of rms, was highly significant for all the three components: their ERD or ERS were stronger in the QM than in the IM. Thus, our results suggest that, while more trials with residual muscle activity can be found in the QM compared to the IM and stronger sensorimotor ERD/ERS can be observed in the QM compared to the IM, the increased residual muscle activity is at least not the main cause of the stronger ERD/ERS in QMs. 

Attempts to make QMs, in general, may lead to such actions that differ from IMs by higher muscle activation, which may go unnoticed for the individual. These actions do not correspond to the strict definition of QMs [1], according to which QMs should not be accompanied by increased EMG, and therefore could be called “quasi-QM” (QQM). However, the only sign of the possible qualitative difference between the QQM and the QM in a strict sense (those that were in the same rms range as IM) was the mild inflection of group trial data CDF at rms value of about 0.5 (Figure 5). Moreover, the contralateral alpha band ERD was almost identical for the QM and QQM (Figure 7, second subplot), which was further supported by the LMM results. 

Most importantly, the movement-related changes in all movement-related sensorimotor EEG components which we identified (contralateral alpha, ipsilateral alpha, contralateral beta) were stronger in QMs than in IMs, *independent* of muscle activation. Thus, if the continuum hypothesis is still true, the causes of the intermediate position of QMs ERD/ERS between IMs and OMs should not be directly linked to the residual muscle activation. 

Our results are very preliminary in what is related to the subjective correlates of the difference between QMs with low and elevated EMG, in other words, between the QMs in a strict sense and the QQMs. It is not unlikely that with more focused questions and higher sample size, subjective correlates of the difference between the QM and the QQM will be found. Here is another evident direction for future QM studies. We are already trying to make a first step in this direction by analyzing detailed reports given by our participants in a free form (the results will be presented in another article [32]. One important observation in that study is that QMs may be much closer to OMs than to IMs in the basic intention, which is to make a real movement both in OMs and QMs (this is also much in line with observations by Nikulin et al. [1]). Therefore, it seems more plausible to consider QMs as a phenomenon which is significantly closer to attempted movements than to the kinesthetic motor imagery. This perspective is interesting to consider, especially from a practical point of view, due to the effectiveness of attempted movements in BCI [3,4]. 

While we expected that the synchronization with rhythmic sounds could lead to a higher rate of trials with increased EMG, this suggestion was not specifically tested (we did not have a control condition with sufficient amounts of self-paced movements, and we could compare our results with the results from the previous studies only using average EMG estimates). More importantly, we cannot be fully sure that the EMG metric we used was sufficiently sensitive to all relevant forms of the residual muscle activation. These issues should be clarified in future studies. 

Another limitation of the study was the use of some manual procedures (selection of the spatial filters and setting the individual frequency bands). We did not expect that a degree of subjectivity in these procedures could introduce a significant bias to the results, because, in almost all cases, the choice of spatial filters was evident, while the effect of small variations in defining the individual frequency bands on ERD/ERS could not be strong. However, in future studies and especially in applications, the automatization of these procedures would be helpful.

In our study, we refrained from analyzing EEG spatial patterns, which would require methodology that is significantly different from what we used. Contrasting the conditions that were most different in their ability to affect the sensorimotor rhythms, as we did, should enable more precise construction of the spatial filters, which then could be applied also to other conditions. However, if the spatial pattern of ERD/ERS was similar between OMs and QMs, but very different between them and IMs, our processing could lead to the artificial suppression of the observed ERD/ERS amplitude in IMs compared to QMs. Based on Nikulin et al.’s [1] observations, we assess the probability to have such a high spatial difference between ERD/ERS in IMs and QMs as very low. Nevertheless, a precise comparison of spatial patterns between IMs and QMs in future studies would be helpful. both to fully rule out this possibility and to better understand the nature of QMs and IMs.

In line with Nikulin et al. [1], but not with Zich et al. [8], we observed higher ERD/ERS effects for the QM compared to the IM. The absence of the difference between QMs and IMs in the Zich et al. study could come from shorter QM training received by their participants (although we cannot fully exclude that, in our case, the need to synchronize movements with sounds could somehow modify the effects). However, such as in the previous studies, participants in our study did not practice for many sessions, which is known to be helpful to obtain stronger ERD with motor imagery based BCIs. It is unclear if QMs will still provide advantages if QMs and IMs would both be practiced long enough, and this also should be investigated. Nevertheless, QM’s ability to provide better ERD/ERS without long training will remain important even if it will not be as beneficial with longer practice, as it is often helpful to have an opportunity to start using a BCI quickly.

Note that in our and in the previous QM studies attempts to make movements with the smallest possible effort, they were compared to motor imagery, which was, actually, the imagination of movements made with normal amplitude. Therefore, our and Nikulin et al.’s [1] observations of higher EEG effects for QMs compared with IMs, and our observation that these effects did not depend on the muscle activation, may have implications for the understanding of how the cerebral cortex supports turning motor intentions into actions: it seems that the intention to make a *real*, not imaginary movement matters more for the ERD/ERS of the sensorimotor rhythms than the intensity characteristics of the movement at low intensity levels (although it remains to be studied if there will be no qualitative difference in ERD/ERS between QMs and small, real movements). Moreover, these observations suggest that low-effort movement attempts (not necessarily QMs) should be considered as a possible instrument of post-stroke rehabilitation, either BCI-assisted or performed without a BCI. Furthermore, they support the suggestion [1] that QMs can be a useful model for developing BCIs based on attempted movements with experiments in healthy participants.

Finally, the ability of QMs to act as a substitute of the attempted movements for eliciting a substantial ERD/ERS, confirmed in this study, opens a way to use them in active BCIs for healthy users, where an unusual experience is an important feature [33]. 

## 5. Conclusions

We found that quasi-movements of thumb abduction were associated with stronger effects on the EEG sensorimotor rhythms than the kinesthetic motor imagery of thumb abduction, and this difference in effects was independent from the difference in the residual activation of the related muscle. The results suggest that quasi-movements, in their brain mechanisms, could be closer to the attempted movements than to the kinesthetic motor imagery. Quasi-movements may appear to be useful in neurorehabilitation, in modeling BCI control with attempted movements in healthy participants, and in creating recreational BCIs for healthy users.

## Figures and Tables

**Figure 1 life-13-00303-f001:**
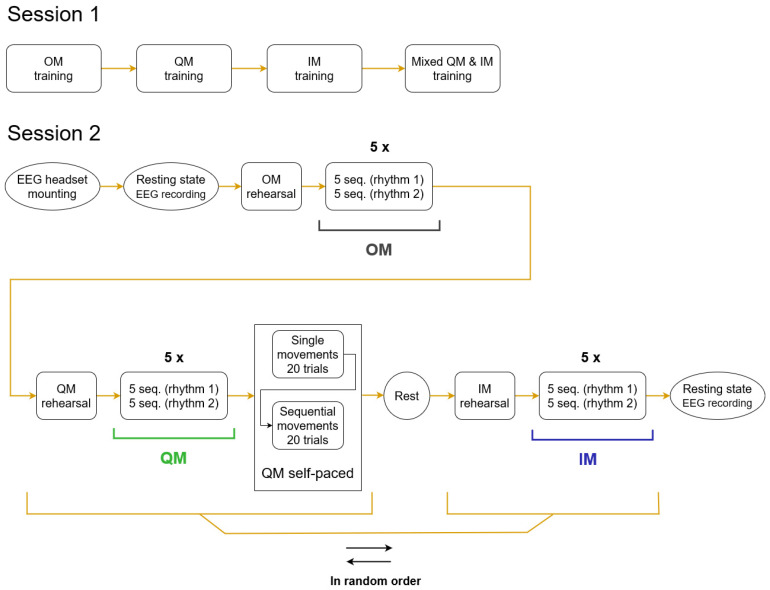
Flowcharts of the two sessions. The EMG and EEG data used in the analysis in the current study were taken from the conditions of OM, QM, IM designated with horizontal black, green and blue lines. Within a condition, five pairs of blocks were presented, each block consisting of five trial sequences (seq.). The order of periodic and aperiodic rhythms in a pair of blocks (rhythm 1, rhythm 2) was fixed for a participant but randomized over the group.

**Figure 2 life-13-00303-f002:**
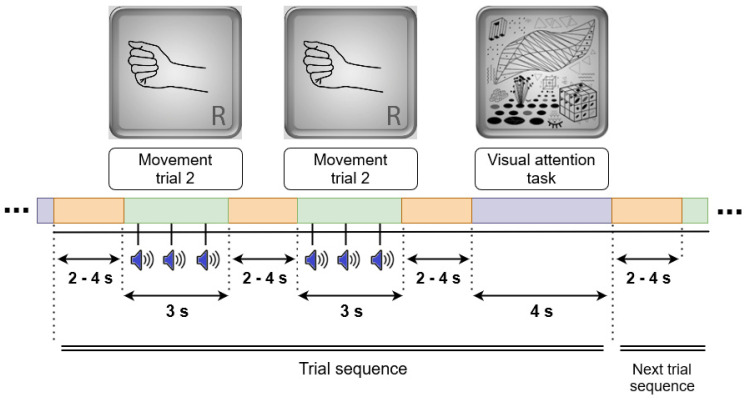
The structure of a trial sequence. Each trial sequence consisted of two movement trials (OM, QM, or IM) and one visual task (reference) trial, with varying length intervals for rest between them. Visual images (top row) were presented to the participants during the related trials.

**Figure 3 life-13-00303-f003:**
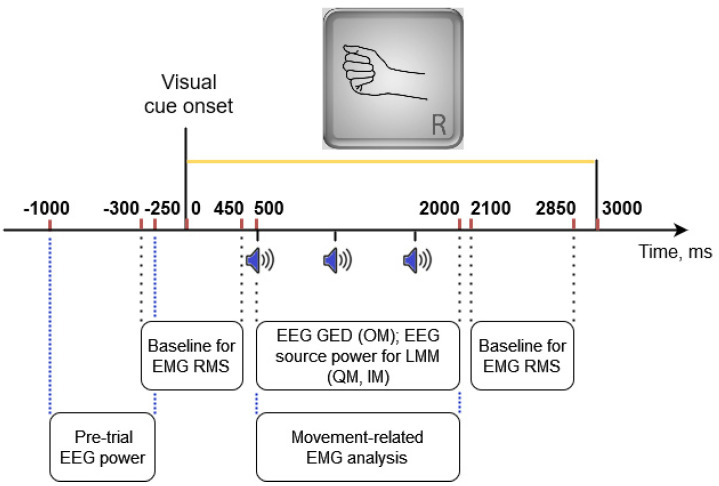
Time intervals from a movement (OM, QM, or IM) trial providing data that were used in different analyses. Sound onsets were at 500, 1100, 1700 ms for the periodic rhythm and at 500, 900, 1700 ms for the aperiodic rhythm.

**Figure 4 life-13-00303-f004:**
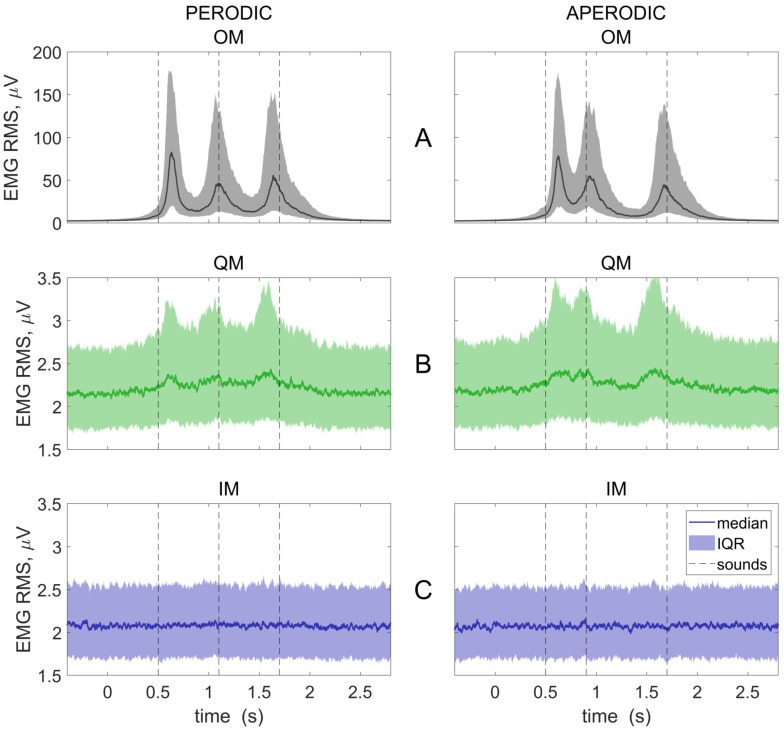
Group averaged EMG RMS time courses for three conditions: OM, overt movements (**A**); QM, quasi-movements (**B**); and IM, imaginary movements (**C**); and two rhythmic patterns (left and right columns). Baseline (−300…450 ms and 2100…2850 ms) was subtracted. Individual trials were collapsed before averaging. The 0 ms corresponds to visual cue onset. Medians and interquartile ranges (IQR) are presented. Vertical lines show the onsets of the sounds, to which the movements should be synchronized.

**Figure 5 life-13-00303-f005:**
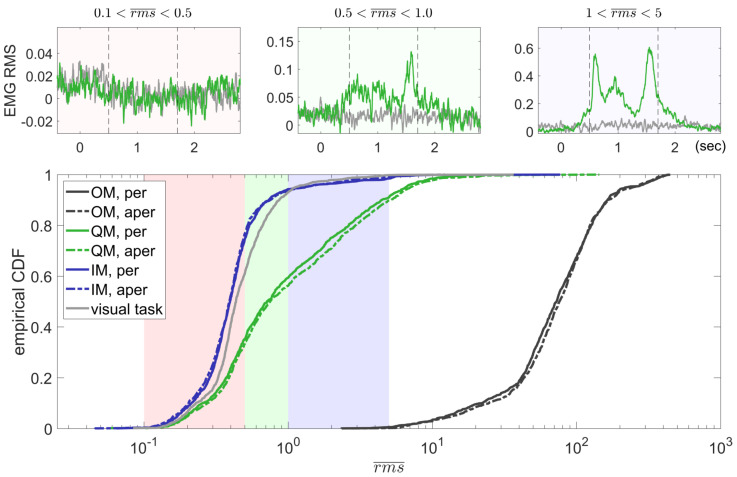
Cumulative distribution functions (CDFs) for the trial $\bar{rms}$ values (95th percentile of the corrected EMG RMS) computed for different conditions. Trial data were collapsed over participants and the rhythm patterns (periodic and aperiodic). Smaller subplots show the time courses of averaged QM and visual task RMS EMG data (normalized by SD) for some $\bar{rms}\;$ ranges (selected for illustrative purposes). Vertical lines in the smaller subplots show the onsets of the 1st and 3rd sounds (the 2nd one differed between periodic and aperiodic rhythm patterns). Here, the visual task data were taken from the QM trial sequences, to make more definitive the comparison between them and the QM data. For the third range, too few visual task trials were found, so the gray line in the third top subplot was drawn for the subset of trials for the joint visual task trial subset, including trials for both second and third ranges.

**Figure 6 life-13-00303-f006:**
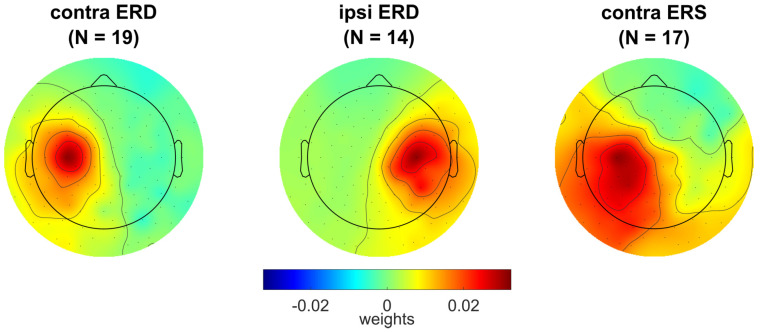
Grand averaged activation scalp topographies for the EEG sensorimotor rhythm components: exhibiting movement-related contralateral ERD in alpha band (*contra ERD*), ipsilateral ERD in alpha band (*ipsi ERD*), and contralateral ERS in beta band (*contra ERS*). For individual topographies of these components, see Appendix A.

**Figure 7 life-13-00303-f007:**
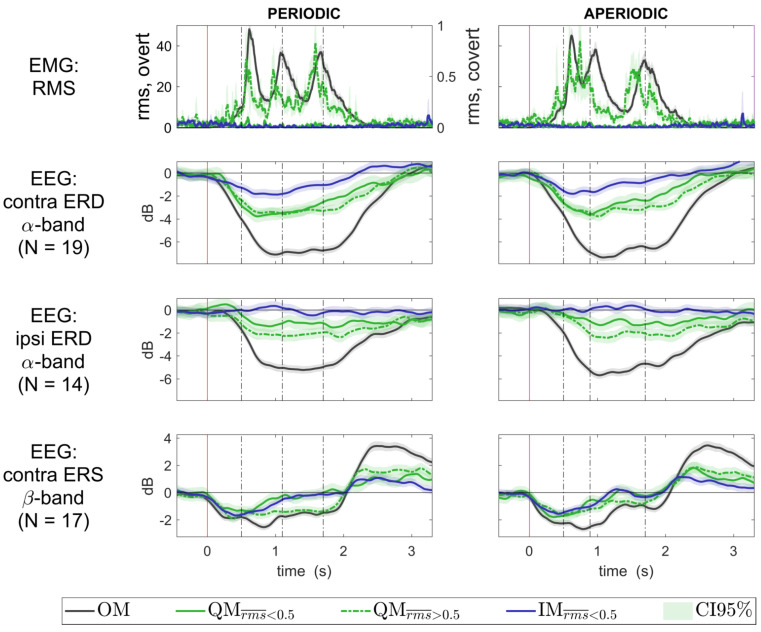
Group averaged time courses of the EMG RMS and the EEG ERD/ERS in all movement conditions. QM data are presented separately for trials with low ($\underline{rms}$ < 0.5) and elevated (0.5 < $\underline{rms}$ < 5.0) EMG. Power values for the components of the EEG sensorimotor rhythms were averaged over the same frequency range (alpha and beta, both defined individually, see Appendix A), which was used previously to define the EEG components. ERD/ERS was quantified as normalized band power of the EEG components, with baseline set at −500…0 ms. Individual trial data were collapsed over the group before averaging. The 0 ms corresponds to the visual cue onset. Vertical lines show the onsets of the sound.

**Figure 8 life-13-00303-f008:**
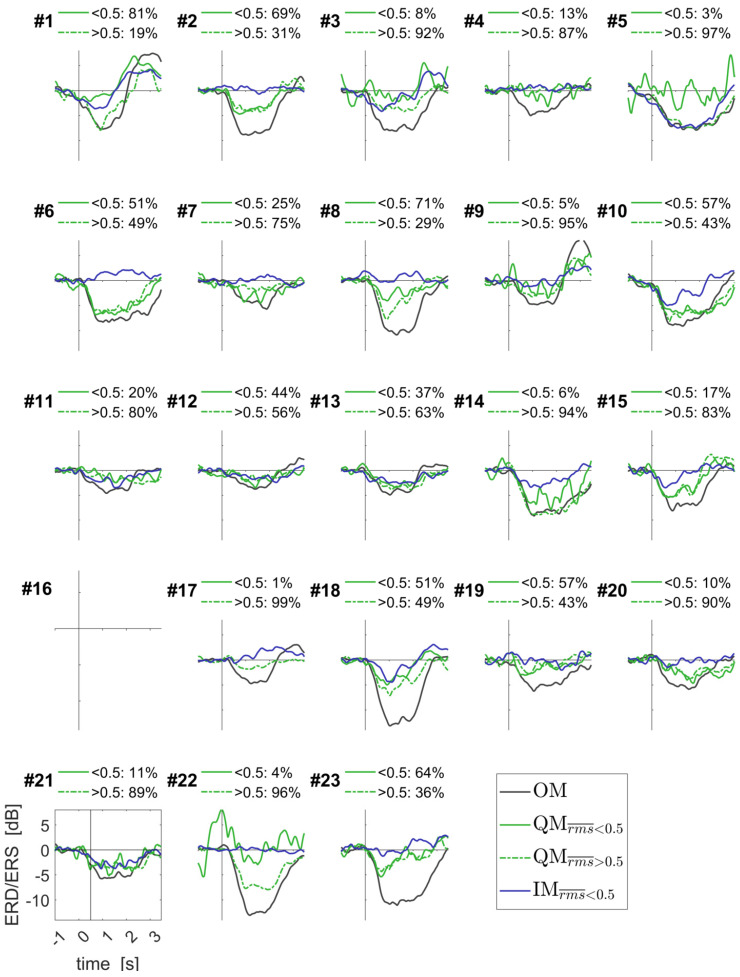
Individually averaged time courses of the ERD/ERS for the contralateral alpha band component of the EEG sensorimotor rhythms. Data for periodic and aperiodic sound rhythms were put together. Individual percentages of trials are shown for QM with low and elevated EMG RMS, as quantified by $\bar{rms}$. See Figure 7 caption for more details and Appendix A for other components.

**Table 1 life-13-00303-t001:** Independent variables used in the linear mixed-effects models.

Var Name	Description	Values
*cond*	Type of covert movement (condition)	quasi, imagery
*rhythm*	Rhythm pattern	periodic, aperiodic
*recN*	Block number	1, 2, …, 13
*trialN*	Trial number within a block	1, 2, …, 10
*prevT*	Type of previous trial	movement, visual
*subj*	Participant ID	“1”, “2”, … “23” ^1^
*peakRMS*	$\mathrm{Peak}\;\mathrm{value}\;\mathrm{of}\;\mathrm{corrected}\;\mathrm{EMG}\;\mathrm{RMS}\;(\bar{rms}$)	Continuous
*prePower*	Pre-movement mean alpha/beta power (−1000…−250 ms)	Continuous

^1^ The number of levels for *subj* depended on the EEG component, as some components could not be found in some participants.

**Table 2 life-13-00303-t002:** Participants’ responses to selected questions in Session 2 and percentage of QM trials with elevated EMG. QM trials with elevated EMG were defined as trial with $\bar{rms}$ > 0.5 (see Methods). Percentage higher than 80% is shown in bold. Here, percentage was computed for trials collapsed over periodic and aperiodic pattern conditions.

Participant ID	“Did You Feel That You Were Moving Your Thumb When You Executed the [QM]?” ^1^	“Did You Feel That You Were Tensing a Muscle When You Executed the [QM]?” ^1,2^	“Were the Actions You Performed in This [the QM] Task Real Rather Than Imaginary?” ^1,3^	QM Trials with Elevated EMG, % of All QM Trials
1	Sometimes	-	No	17
2	Yes	-	Yes	30
3	Yes	-	More real	**82**
4	Sometimes	-	Yes	73
5	Yes	-	More real	**98**
6	No	-	Between real and imaginary	38
7	No	-	Between real and imaginary	58
8	Yes	-	Yes	31
9	Yes	No	Yes	72
10	No	Yes	Yes	24
11	Sometimes	Sometimes	Yes	79
12	No	Yes	Yes	49
13	Sometimes	Sometimes	Yes	56
14	No	Yes	No	**85**
15	No	Yes	Yes	73
16	No	No	More real	6
17	No	Yes	Yes	**99**
18	Yes	Yes	Yes	48
19	No	Yes	Yes	22
20	No	No	More imaginary	**86**
21	No	Sometimes	Yes	54
22	No	Yes	Yes	**95**
23	Yes	Yes	Yes	28
N of “Yes”	7	9	15	
N of “No”	12	3	2	

^1^ For quasi-movements, the word “micromovements” was actually used for quasi-movements (see the instructions to participants in the Methods section). ^2^ This question was not asked to the participants 1–8. ^3^ This question was actually asked in the form: “*To what extent were the actions you performed in this task imaginative or real?*” Here, the question and the responses are slightly reformulated (keeping the sense of the responses unchanged) to make the table more easy-to-read.

## Data Availability

The data presented in this study are available upon reasonable request from A.N.V. or S.L.S.

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
