# Peer review of "Quasi-Movements and “Quasi-Quasi-Movements”: Does Residual Muscle Activation Matter?"

_life, 2023, doi:10.3390/life13020303_

Round 1

Reviewer 1 Report

In the present study the authors scrutinized further EMG and EEG correlates of quasi-movements (QM), i.e. movements which are minimized to such an extent that they are not present when assessed objectively with EMG measures. QM represent a phenomenon that is much closer to the attempted movements  than motor imagery since in the latter there is no intention to perform a real movement. Therefore investigation of QM and their neuronal correlates is important for the understanding of efferent processes during movement performance and for developing novel behavioral strategies in the context of BCI research. A focus in this study was on the effect of residual EMG during QM and whether it can affect central neuronal processing manifested in ERD/ERS of sensorimotor oscillations. The study is well conceived and is performed on a good technical level. I have the following comments:

For the GED spatial filtering the authors used only visual vs OM contrast. Why not to use a separate GED for QM and IM as well? One can contract them against a visual condition. Obtained patterns (inverse of the spatial filter) would allow a comparison of spatial activations between the three conditions. I understand that the idea was to use the same filters for QM and MI but it can be that they had differences in spatial activation. Alternatively, the authors can also explain in more details why their approach is still sufficient. 

The authors write: “This contradicts the understanding of QM as a part of a continuum between IM and overt movements (OM): if the continuum hypothesis was true, muscle activation in QM would likely be highly correlated with the EEG effects of the action.” This is an interesting observation. However, I would like to offer the following considerations:

1.       QM ERD strength is indeed intermediate between OM and IM.

2.       There might be a scaling with neuronal activity w.r.t to the residual EMG but the sensitivity of EEG is not sufficient to detect it.

Altogether I do not see why continuity hypothesis is not supported by the data obtained by the authors. Yet I like also the suggestion that QM are closer to attempted movements.

And I absolutely agree with the authors that QM have a potential for rehabilitation purposes.    

In Figure 4 please indicate units on Y-axis (microvolts I assume but if so why only 0.3 for QM?). I would recommend using here absolute, not normalized values.

On page 10, step 10. Please clarify whether your patterns were mostly radial or tangential.

Where was the reference for the EEG acquisition (there is only info about the ground at Fpz).

Page 13. The authors write: “… In general, the participants' confidence in the reality of the CM                increased during the second session,” Do you mean QM, not CM?

Author Response

We are very thankful to Reviewer 1 for careful reviewing the manuscript and providing thoughtful and helpful suggestions. Below are our responses, each of them describes a related change made in the manuscript using the “Track Changes” function of MS Word:

For the GED spatial filtering the authors used only visual vs OM contrast. Why not to use a separate GED for QM and IM as well? One can contract them against a visual condition. Obtained patterns (inverse of the spatial filter) would allow a comparison of spatial activations between the three conditions. I understand that the idea was to use the same filters for QM and MI but it can be that they had differences in spatial activation. Alternatively, the authors can also explain in more details why their approach is still sufficient. 

- To justify our approach and also to acknowledge its limitations, we added the following paragraph to Discussion:

In our study, we refrained from analyzing EEG spatial patterns, which would require methodology that is significantly different from what we used. Contrasting the conditions that were most different in their ability to affect the sensorimotor rhythms, as we did, should enable more precise construction of the spatial filters, which then could be applied also to other conditions. However, if the spatial pattern of ERD/ERS was similar between OM and QM but very different between them and IM, our processing could lead to artificial suppression of the observed ERD/ERS amplitude in IM compared to QM. Based on Nikulin et al. [1] observations, we assess the probability to have such a high spatial difference between ERD/ERS in IM and QM as very low. Nevertheless, a precise comparison of spatial patterns between IM and QM in future studies would be helpful both to fully rule out this possibility and to better understand the nature of QM and IM.

To clarify the terminology for the readers who are familiar with the wide use of term “CSP” in the BCI literature but are not aware of its relation to GED we slightly updated the sentence “Spatial filtering was performed using generalized eigendecomposition (GED) ….” in Methods, which now has the following form:

Spatial filtering was performed using generalized eigendecomposition (GED) on covariance matrices (also known as Common Spatial Pattern, CSP, in the BCI literature), which was shown to effectively extract functionally relevant components with different spatial patterns from the EEG [14–16].

The authors write: “This contradicts the understanding of QM as a part of a continuum between IM and overt movements (OM): if the continuum hypothesis was true, muscle activation in QM would likely be highly correlated with the EEG effects of the action.” This is an interesting observation. However, I would like to offer the following considerations:

  1.   QM ERD strength is indeed intermediate between OM and IM.
  2.   There might be a scaling with neuronal activity w.r.t to the residual EMG but the sensitivity of EEG is not sufficient to detect it.

Altogether I do not see why continuity hypothesis is not supported by the data obtained by the authors. 

- We agree with this argument, and made several related changes to Discussion to provide a more accurate view on the possible implications from our results for the continuity hypothesis and some other related issues:

(1) The second sentence in the first paragraph of Discussion was replaced with a few different sentences: 

Previous studies of quasi-movements (QM) [1,8] and theoretical reflections on them in the literature (e.g., [31]) suggested that this phenomenon should be considered as a part of a continuum between overt and imaginary movements. Results of the current study call for a somewhat different view on the QM.

was changed to:

Previous studies of quasi-movements (QM) [1,8] and theoretical reflections on them in the literature (e.g., [31]) suggested that this phenomenon should be considered as a part of a continuum between overt and imaginary movements. One of its manifestations could be the intermediate strength of sensorimotor rhythm modulation in QM, positioned between IM and OM, which was confirmed in the current study. Such intermediate position of QM ERD could be related to intermediate levels of activation of the efferent system or other underlying mechanisms that manifest themselves in the variations of the residual muscle activation. The current study provides strong arguments against such a view. 

(2) Second sentence of the 4th paragraph was changed in the following way:

old version:

Most importantly, the movement-related changes in all movement-related sensorimotor EEG components which we identified (contralateral alpha, ipsilateral alpha, contralateral beta) were stronger in QM than in IM independently of muscle activation. This contradicts the understanding of QM as a part of a continuum between IM and overt movements (OM): if the continuum hypothesis was true, muscle activation in QM would likely be highly correlated with the EEG effects of the action. 

new version:

Most importantly, the movement-related changes in all movement-related sensorimotor EEG components which we identified (contralateral alpha, ipsilateral alpha, contralateral beta) were stronger in QM than in IM independently of muscle activation. Thus, if the continuum hypothesis is still true, the causes of the intermediate position of QM ERD/ERS between IM and OM should not be directly linked to the residual muscle activation. 

In addition, to make the logical structure of the text more clear, the content of the 5th paragraph of Discussion was moved to the end of the next paragraph (which is no. 5 in the current version of Discussion), where the last sentence was this:

We are already trying to make a first step in this direction by analyzing detailed reports given by our participants in a free form (the results will be presented in another article).

so now is appended in the following way:

We are already trying to make a first step in this direction by analyzing detailed reports given by our participants in a free form (the results will be presented in another article – Yashin et al., in prep.). One important observation in that study is that the QM may be much closer to OM than to IM in the basic intention, which is to make a real movement both in OM and QM (this is also much in line with observations by Nikulin et al. [1]). Therefore, it seems more plausible to consider QM as a phenomenon which is significantly closer to attempted movements than to the kinesthetic motor imagery. This perspective is interesting to consider especially from a practical point of view, due to the effectiveness of attempted movements in BCI [3,4]. 

In Figure 4 please indicate units on Y-axis (microvolts I assume but if so why only 0.3 for QM?). I would recommend using here absolute, not normalized values.

- In the original version, the values were normalized by SD (which we actually forgot to mention in the caption). We now have replaced the figure with a new version where units are microvolts.  

On page 10, step 10. Please clarify whether your patterns were mostly radial or tangential.

- We added the following sentence to step 10: 

Almost all selected alpha band components were consistent with the radial dipole orientation, while both radial and tangential configurations were seen among beta band components (Figure S2).

Where was the reference for the EEG acquisition (there is only info about the ground at Fpz).

- To clarify this issue, we replaced the sentence “The ground electrode was placed at the Fpz position.” in the first paragraph of subsection 2.7. Signal acquisition by the following two sentences:

The EEG was obtained and stored as single-ended signals with the ground electrode at the Fpz position. As we did not analyze the EEG channelwise beyond bad channel detection and interpolation (the main EEG analysis was run solely on components), we did not use a reference channel.

To clarify the recording setup also for the EMG, we added the following sentence to the same paragraph: 

The signal was re-computed to a single bipolar EMG channel both for online visualization and for the processing.

Page 13. The authors write: “… In general, the participants' confidence in the reality of the CM            increased during the second session,” Do you mean QM, not CM?

- yes, this was a typo, we now replaced CM with QM

Reviewer 2 Report

This study presents new evidence on the differences between brain mechanisms underlying different types of movements (overt, Imaginary, quasi-movements), based on the study of the EMG signals and the ERD/ERS derived from EEG measurements in healthy participants. The study is well-designed and can have important implications for both basic understanding of brain functioning and the development of technological applications based on Brain Computer Interfaces, Neurofeedback and/or neurorehabilitation. I do not have any major concern although I do have several minor comments that should be addressed before accepting the manuscript for publication.

1.- Title/Introduction: As the term quasi-quasi-movements is not well-known and it seems more like a definition of the authors for the type of QM they are studying in their experiment here, I think it should be introduced and explained in the introduction, so it is easier for the reader to quickly understand the title of the manuscript. As an alternative, I think it would be better to set the title around QM and IM which are general well-known terms (i.e. Quasi-movement and imaginary movements: does residual muscle…), and I think it does not change the attractiveness and the main idea of the study (and it matches better with the conclusions in the manuscript). This latter option would avoid large changes in the introduction and the difficulty to introduce the new term “QQM” that is indeed better to understand after going through the Method section.

2.- Introduction: It is not clear if the last paragraph (line 124) refers to previous studies by the authors or by the results presented in the current manuscript. If it is the former it should be better referenced and if it is the latter, I would suggest to leave those comments to the Discussion.

3.- Figures can be improved for better readability.

- The space in Figure 1 should be better used in order to have larger fonts for every box in the scheme. Please, also check if the box that represents the rehearsal before the IM experiment should not say IM rehearsal instead of QM rehearsal. A final suggestion (optional) would be to use the words “5 blocks x” above the boxes representing the main experiment for OM, QM and IM to make it clearer and simpler, and better matching the caption.

- Figure 2 and especially Figure 3 can also be resized to increase visibility of the attention task and readability of the boxes.

- Figure 4 to 8. Fonts of the letters and numbers in axes are too small, making it difficult to correctly see and interpret the charts. Applies also to Figures S1, S4 and S5.

- Figure S3. Showing the ERD and ERS in the spectrograms might benefit of using a non-symmetric colorbar, such that maximum ERS amplitudes can reach the red color. This also makes easier the comparison among the three components.

- Figure 7. The caption is not correct. It seems a copy-paste from caption of Figure 6. Figure 7 shows time courses of EMG and ERD/ERS, and not scalp topographies.

4.- An important issue is that sometimes in the manuscript the EEG components defined by spatial filters are referred to as EEG sources. It is very important that authors refrain to explain or discuss results in terms of localization of sources, as they are always working at the level of scalp topographies. Due to the well-known volume conduction effects, it is not straightforward to claim that the different extracted components are completely separated (e.g. line 199, should not be “could not affect” but better something as “are assumed/expected to not largely affect…”). I also suggest to change all instances of “EEG sources” by “EEG components”, as used in section 3.2 and 3.3 (lines 665, 666, 671, 695, 699, 700, 705-707, 709, 716, 717), as well as to remove the sentence in lines 655 to 657.

5.- Authors have decided to call alpha and beta rhythm to sensorimotor rhythms that have been largely defined as a different rhythm from the occipital alpha activity. For instance, books such as Electroencephalography, 5th Ed., by Nierdemeryer and Lopes da Silva, and the pioneering work on ERD/ERS of sensorimotor rhythms by Pfurtscheller, and a lot of current literature generally use Mu rhythm or SMR (sensorimotor rhythm). I think these differences and the decision to use such terminology should be clearly stated in the Introduction or in the Methods in order to avoid confusion in younger readers and to make clearer why the components were found to have central topographical distributions.

6.- From the last paragraph of section 2.4.3, the reader can understand (at least I do) that some feedback was given by the experimenter to the participant in the IM training, in order to decrease the EMG amplitude. Might this be the reason why stable IM was faster to reach than QM? Why the same feedback was not given in the QM practice? Might this also be why participants showed more EMG in QM than in IM? I think this potential bias should be acknowledge and discussed by the authors or give the arguments why their main result is not just a consequence of the experimental manipulation in the training phase.

7.- Please report the reference electrode in section 2.7. This usually has implications for the activation topographies of EEG components.

8.- Line 376. Please report the method used to compute the Power Spectral Density and give the full name for the acronym (PSD).

9.- In general, I think it would be very useful for readers (especially those interested in stepping on the results presented here) to have at least a paragraph in the discussion for stating potential methodological limitations of the study, given the many steps included. For instance, the way the spectral bands of interest were defined, or the selection of components based on visual inspection of their spatial distribution to ensure it represents sensorimotor activity. These options could have implications for the feasibility of claimed BCI applications, which would be difficult to achieve using the same methodological procedure followed in this study.

10.- Section 3.2. It seems to me that there are some inconsistencies between the number of participants reported in the text and those in the figures. For instance, for describing Figure 6 the main text explains that there were 23, 16 and 20 subjects in each of the three EEG components, but the Figure shows the topographies averaged across a different number of subjects. It seems that the figure was indeed computed after removing other three subjects (participants 5,17,22), but this exclusion is explained later as applied only to subsequent analysis. Please check and correct the numbers if necessary.

11.- Paragraph in lines 779 to 783 might benefit by including a discussion about the LMM results on the absent of significant effects of the type of rhythmic sounds on the ERD/ERS components.

 Typos

Although the paper is well written and generally easy-to-follow, I found some typos and I list some of them here to help authors with the revision.

- Simple Summary. Page 1, line 15. Replace “no” by “or”.

- Introduction. Page 2, line 81. You should remove the word “not”.

- Line 136. Correct “experiment” to “experimental”.

- Line 221. Correct “proposes” to “purposes”.

- Lines 257 and 264. Use the plural “participants” so it matches the possessive “their”.

- Several times the acronym MI is used instead of IM. Please check and correct all instances. Some are found in lines 274, 280, 286, 291, 292, 321, 541.

- Line 276. Correct to “Improving understanding of the difference between QM and IM”.

- Line 299. The last phrase “…data are beyond the scope…” seems odd. I suggest to rephrase it as “…but the analysis of these data is beyond the scope… ” or “…but these data were not analyses for the purposes of this paper.”

- Line 326. Remove the word “a”.

- Line 409. The reference 17 is not Olive and Hawkins, but Olive only. Please correct.

- Line 543. CM should be IM right? Otherwise, define it.

- Legend of Table 2. Check the question in the foot page note 2. I think this should be the question related to tension the muscle and not regarding the feeling of actions as real or imaginary.

- Lines 627-628. The sentence is not easy to understand. Pleas use “15 out of 23” and probably better to rephrase it to make it clearer.

- Line 635. Replace “another two” by “the other two”.

- Line 692. Correct “capture” to “caption”. By the way, it sends the reader to caption of Figure 7 which wrongly refers to scalp topographies instead of time courses.

- Lines 737 and 739. Correct “feeled” to “felt”.

- Line 783. Change “one more” to “another”.

- Line 816-817. Please check the sense of this sentence. It seems to me that it should read “…closer to the attempted movements than the kinesthetic motor imagery”. The current use of the phrase “than to the kinesthetic…” implies that the QM are closer to OM than to IM, which is not the case according to results on EMG rms (Figure 5) and ERD/ERS (Figure 7).

Author Response

We are very thankful to Reviewer 2 for careful reviewing the manuscript and providing detailed thoughtful and helpful suggestions. Below are our responses, each of them describes a related change made in the manuscript using the “Track Changes” function of MS Word:

1.- Title/Introduction: As the term quasi-quasi-movements is not well-known and it seems more like a definition of the authors for the type of QM they are studying in their experiment here, I think it should be introduced and explained in the introduction, so it is easier for the reader to quickly understand the title of the manuscript. As an alternative, I think it would be better to set the title around QM and IM which are general well-known terms (i.e. Quasi-movement and imaginary movements: does residual muscle…), and I think it does not change the attractiveness and the main idea of the study (and it matches better with the conclusions in the manuscript). This latter option would avoid large changes in the introduction and the difficulty to introduce the new term “QQM” that is indeed better to understand after going through the Method section.

- We agree that the QQM should be defined in Introduction if it is in the title. This phenomenon was actually discussed in the Introduction in the original version, but without using the term. To keep the term in the title, we changed one sentence in the Introduction in such a way that now it defines it:

If the procedures they used for EMG analysis underestimated the rate of QM trials with elevated task-related EMG, which we will also call “quasi-quasi-movements” (QQM), this could, in turn, cause underestimation of the correlation between the EMG and the EEG effects.

2.- Introduction: It is not clear if the last paragraph (line 124) refers to previous studies by the authors or by the results presented in the current manuscript. If it is the former it should be better referenced and if it is the latter, I would suggest to leave those comments to the Discussion.

- According to the journal instructions for authors, we need to “highlight the main conclusions” at the end of Introduction. However, description of the study objectives between the word “here” and these conclusions is rather lengthy, and we agree that it was not clear enough if the conclusions are related to this or other study. To avoid such confusion, we now added words “in this study” before the conclusions.

3.- Figures can be improved for better readability.

- The space in Figure 1 should be better used in order to have larger fonts for every box in the scheme. Please, also check if the box that represents the rehearsal before the IM experiment should not say IM rehearsal instead of QM rehearsal. A final suggestion (optional) would be to use the words “5 blocks x” above the boxes representing the main experiment for OM, QM and IM to make it clearer and simpler, and better matching the caption.

- The proposed changes have been made.

- Figure 2 and especially Figure 3 can also be resized to increase visibility of the attention task and readability of the boxes.

- Figure sizes have been increased.

- Figure 4 to 8. Fonts of the letters and numbers in axes are too small, making it difficult to correctly see and interpret the charts. Applies also to Figures S1, S4 and S5.

- Fonts have been enlarged.

- Figure S3. Showing the ERD and ERS in the spectrograms might benefit of using a non-symmetric colorbar, such that maximum ERS amplitudes can reach the red color. This also makes easier the comparison among the three components.

- We changed the color scale following this suggestion.

- Figure 7. The caption is not correct. It seems a copy-paste from caption of Figure 6. Figure 7 shows time courses of EMG and ERD/ERS, and not scalp topographies.

- Corrected.

4.- An important issue is that sometimes in the manuscript the EEG components defined by spatial filters are referred to as EEG sources. It is very important that authors refrain to explain or discuss results in terms of localization of sources, as they are always working at the level of scalp topographies. Due to the well-known volume conduction effects, it is not straightforward to claim that the different extracted components are completely separated (e.g. line 199, should not be “could not affect” but better something as “are assumed/expected to not largely affect…”). I also suggest to change all instances of “EEG sources” by “EEG components”, as used in section 3.2 and 3.3 (lines 665, 666, 671, 695, 699, 700, 705-707, 709, 716, 717), as well as to remove the sentence in lines 655 to 657.

- We agree that it is important not to mix EEG sources and components. We made the required changes at the instances listed by the Reviewer.

5.- Authors have decided to call alpha and beta rhythm to sensorimotor rhythms that have been largely defined as a different rhythm from the occipital alpha activity. For instance, books such as Electroencephalography, 5th Ed., by Nierdemeryer and Lopes da Silva, and the pioneering work on ERD/ERS of sensorimotor rhythms by Pfurtscheller, and a lot of current literature generally use Mu rhythm or SMR (sensorimotor rhythm). I think these differences and the decision to use such terminology should be clearly stated in the Introduction or in the Methods in order to avoid confusion in younger readers and to make clearer why the components were found to have central topographical distributions.

- We actually tried to follow the standard terminology, but it seems that in some cases the wording we used could indeed make an impression that we mean “alpha rhythms” and “beta rhythms” instead of alpha and beta band components of the sensorimotor rhythms. We now checked all instances of mentioning alpha and beta and made the necessary corrections, to ensure full correspondence to the standard terminology.

6.- From the last paragraph of section 2.4.3, the reader can understand (at least I do) that some feedback was given by the experimenter to the participant in the IM training, in order to decrease the EMG amplitude. Might this be the reason why stable IM was faster to reach than QM? Why the same feedback was not given in the QM practice? Might this also be why participants showed more EMG in QM than in IM? I think this potential bias should be acknowledge and discussed by the authors or give the arguments why their main result is not just a consequence of the experimental manipulation in the training phase.

- As we stated in the description of the QM practice (section 2.4.2), the feedback was also provided in this practice, moreover, this feedback was the essence of this practice (following the original method by Nikulin et al.). To make this more clear, we added words “Like in QM practice,” before mentioning the feedback for IM in section 2.4.3.

7.- Please report the reference electrode in section 2.7. This usually has implications for the activation topographies of EEG components.

- We added the following clarification to the first paragraph in subsection 2.7. Signal acquisition:

The EEG was obtained and stored as single-ended signals. As we did not analyze the EEG channelwise beyond bad channel detection and interpolation (the main EEG analysis was run solely on components), we did not use a reference channel.

To clarify the recording setup also for the EMG, we added the following sentence to the same paragraph: 

The signal was re-computed to a single bipolar EMG channel both for online visualization and for the processing.

8.- Line 376. Please report the method used to compute the Power Spectral Density and give the full name for the acronym (PSD).

- We replaced the words “channel PSD were computed” with the following text:

channel power spectral density was computed using Welch's method (using Matlab pspectrum function with default parameters)

9.- In general, I think it would be very useful for readers (especially those interested in stepping on the results presented here) to have at least a paragraph in the discussion for stating potential methodological limitations of the study, given the many steps included. For instance, the way the spectral bands of interest were defined, or the selection of components based on visual inspection of their spatial distribution to ensure it represents sensorimotor activity. These options could have implications for the feasibility of claimed BCI applications, which would be difficult to achieve using the same methodological procedure followed in this study.

– We already discussed some limitations in the original version of the article. In particular, we acknowledged that we did not compare sound-synchronized movements with self-paced movements. Now we appended the related paragraph (started from “While we expected that synchronization with rhythmic sounds …”) with the following sentence:

More importantly, we cannot be fully sure that the EMG metric we used was sufficiently sensitive to all relevant forms of residual muscle activation.

and added, as the next paragraph, the following text:

Another limitation of the study was the use of some manual procedures (selection of the spatial filters and setting the individual frequency bands). We did not expect that a degree of subjectivity in these procedures could introduce a significant bias to the results, because almost in all cases the choice of spatial filters was evident, while the effect of small variations in defining the individual frequency bands on ERD/ERS could not be strong. However, in future studies and especially in applications, automatization of these procedures would be helpful. 

After it, we added another new paragraph starting from “In our study, we refrained from analyzing EEG spatial patterns …” in response to R1 request, where we discuss some additional limitations.

10.- Section 3.2. It seems to me that there are some inconsistencies between the number of participants reported in the text and those in the figures. For instance, for describing Figure 6 the main text explains that there were 23, 16 and 20 subjects in each of the three EEG components, but the Figure shows the topographies averaged across a different number of subjects. It seems that the figure was indeed computed after removing other three subjects (participants 5,17,22), but this exclusion is explained later as applied only to subsequent analysis. Please check and correct the numbers if necessary.

- Indeed, this exclusion affected Figure 6 as well. Therefore, we moved the sentence explaining the exclusion (“We excluded from further analysis…”) up, before the first reference to Figure 6. We also corrected the number of participants whose beta ERS components were used to draw the averaged topography (Figure 6, from 20 to 17) and added the numbers of participants to the names of components in Figure 7, where they were missed. The number of participants, however, could be higher in the figures that represented individual EEG components, because they could be of interest for some readers even if the data were not analyzed due to insufficient number of low EMG trials.

11.- Paragraph in lines 779 to 783 might benefit by including a discussion about the LMM results on the absent of significant effects of the type of rhythmic sounds on the ERD/ERS components.

- The absence of effects from the sound sequence type is of minor importance for this study, so we only mention the lack of significant effect, to save space (given that the paper is already relatively long). We are going to study sequence type effects in our further analyses, more focused on applications, such as single-trial classification for BCIs. 

Typos

Although the paper is well written and generally easy-to-follow, I found some typos and I list some of them here to help authors with the revision.

- We made the proposed corrections, except for the following two cases which we treated in different ways:

- Line 409. The reference 17 is not Olive and Hawkins, but Olive only. Please correct. 

The reference is indeed Olive, but the method is known as Olive-Hawkins method. We corrected wording to make the sentence less ambiguous.

- Line 816-817. Please check the sense of this sentence. It seems to me that it should read “…closer to the attempted movements than the kinesthetic motor imagery”. The current use of the phrase “than to the kinesthetic…” implies that the QM are closer to OM than to IM, which is not the case according to results on EMG rms (Figure 5) and ERD/ERS (Figure 7).

We do mean that QM are closer to attempted movements than to IM. Note that attempted movements are not OM, they are just attempts, not actual movements. The conclusion is made based not only on ERD analysis but also based on the analysis of intentions, which we now highlight in the updated (according to request by R1) part of Discussion (“the QM may be much closer to OM than to IM in the basic intention, which is to make a real movement both in OM and QM (this is also much in line with observations by Nikulin et al. [1]). Therefore, it seems more plausible to consider QM as a phenomenon which is significantly closer to attempted movements than to the kinesthetic motor imagery”).